# Fate of Horizontal-Gene-Transfer Markers and Beta-Lactamase Genes during Thermophilic Composting of Human Excreta

**DOI:** 10.3390/microorganisms11020308

**Published:** 2023-01-24

**Authors:** Katharina A. Werner, Lara Feyen, Tobias Hübner, Nicolas Brüggemann, Katharina Prost, Elisabeth Grohmann

**Affiliations:** 1Faculty of Life Sciences and Technology, Department of Microbiology, Berliner Hochschule für Technik, 13347 Berlin, Germany; 2Department of Environmental Microbiology, Helmholtz Centre for Environmental Research GmbH—Umweltforschungszentrum Leipzig (UFZ), 04318 Leipzig, Germany; 3Institute of Bio- and Geosciences (Agrosphere, IBG-3), Forschungszentrum Jülich, 52425 Jülich, Germany

**Keywords:** ecological sanitation, compost, antibiotic resistance, horizontal gene transfer, qPCR

## Abstract

Thermophilic composting is a suitable treatment for the recycling of organic wastes for agriculture. However, using human excreta as feedstock for composting raises concerns about antibiotic resistances. We analyzed samples from the start and end of a thermophilic composting trial of human excreta, together with green cuttings and straw, with and without biochar. Beta-lactamase genes *bla_CTX-M_*, *bla_IMP_*, and *bla_TEM_* conferring resistance to broad-spectrum beta-lactam antibiotics, as well as horizontal gene transfer marker genes, *intI1* and *korB*, were quantified using qPCR. We found low concentrations of the beta-lactamase genes in all samples, with non-significant mean decreases in *bla_CTX-M_* and *bla_TEM_* copy numbers and a mean increase in *bla_IMP_* copy numbers. The decrease in both *intI1* and *korB* genes from start to end of composting indicated that thermophilic composting can decrease the horizontal spread of resistance genes. Thus, thermophilic composting can be a suitable treatment for the recycling of human excreta.

## 1. Introduction

Human excrements contain large amounts of nutrients required for plant nutrition, such as phosphorus and nitrogen [1], which remain mostly unused in countries using water-based sanitation. Ecological sanitation via thermophilic composting can serve as a treatment to convert human excreta into a humus-rich fertilizer [2]. Thermophilic composting is an aerobic and exothermic degradation process of organic material mediated by microorganisms. A first mesophilic phase with temperatures between ambient and 45 °C is followed by a thermophilic phase with temperatures above 45 °C, a second mesophilic, and finally a cooling or maturation phase. Predominant microbial species change over the process [3,4]. Biochar as an amendment to compost was shown to improve aeration and increase composting temperatures and the duration of the thermophilic phase [5,6].

Pathogens are usually not adapted to the high temperatures during the thermophilic phase of composting, which leads to their inactivation [4,7,8]. One of the concerns with the use of human excreta in agriculture are antibiotic resistances that could be introduced into the food chain and thereby promote the antibiotic crisis. The accumulation of multiple resistances with a lack of treatment options is a public health burden, leading to an estimated minimum of 700,000 deaths per year [9,10]. There are indications that clinically relevant resistances may be transferred back from environmental to clinical bacteria [11]. This highlights the importance of preventing the contamination of the environment with antibiotic resistance genes (ARGs). Thermophilic composting is increasingly analyzed regarding its impact on ARGs. The decrease of genes conferring resistance to different antibiotic classes has been described. Nevertheless, the efficiency varies in different studies, and some ARGs were found to show a tendency to persist or increase [12,13,14,15,16,17,18]. Biochar amendments to compost can augment the decrease in ARG levels during thermophilic composting of manure, but studies are scarce [16,19]. Li and colleagues showed that the majority of ARGs analyzed were significantly less abundant in chicken manure compost containing 5–20% biochar [19]. Cui et al. assessed the impact of different biochars on chicken manure composting. Interestingly, different trends were observed, with mushroom biochar enhancing the removal of ARGs and rice straw biochar decreasing the removal rate, as compared with the control treatment [20].

Since their discovery, beta-lactam resistances have been shown to rise in Europe [21]. Especially challenging are carbapenemases and extended-spectrum beta-lactamases (ESBL). The first are active against carbapenems, broad-spectrum beta-lactam antibiotics that are usually used for the treatment of infections with multidrug-resistant pathogens. Rising abundances increase the threat of a lack of treatment options [22]. Carbapenemases are found within the Ambler classes A (IMI, SME, NMC, KPC, GES), B (e.g., IMP, VIM, SPM, NDM), and D (e.g., OXA-23, OXA-48) [23]. The IMP-enzymes (imipenem-resistant *Pseudomonas*-type carbapenemase) have been known since 1991, when they were first described in *Pseudomonas aeruginosa* [24]. The corresponding *bla_IMP_* genes are usually encoded on class 1 integrons and have spread to several clinically relevant bacteria, such as *Enterobacteriaceae* and *Acinetobacter* species [22,23,25,26,27]. *Enterobacteriaceae*, including carbapenemase-producing ones, have been identified in many environments, including rivers, estuaries and coastal waters [28,29], marine salterns [30], urban topsoils and dust aerosols [31], or edible vegetables [32,33]. *Acinetobacter* species are likewise ubiquitous in the environment, such as aqueous and soil habitats, food and animals [34,35].

ESBLs can degrade penicillins, third-generation cephalosporins and monobactams. Among nosocomial infections, ESBL-producing *Enterobacteriaceae* are highly prevalent, and associated with more severe clinical courses of a disease or mortality, as compared with non-ESBL nosocomial infections [36,37,38]. Most ESBLs belong to Ambler class A. However, some OXA-variants (class D) inactivate third-generation cephalosporins, which additionally categorizes them as ESBLs. TEM-1 (Temoneira) was the first described beta-lactamase encoded on a plasmid [23,39,40]; whereas TEM-1 is a penicillinase, most of the 183 TEM variants described so far belong to ESBLs. TEM is highly prevalent, and associated with clinical *Enterobacteriaceae*, *Neisseria gonorrhoeae* and to a lesser extent *P.s aeruginosa* [23,40]. First identified in the 1980s, CTX-M (cefotaximase from Munich) belongs to the most prevalent ESBLs nowadays, and is even more prevalent than TEM and SHV-5. The *bla_CTX-M_* genes are associated with plasmids, transposons and insertion sequences and are found in many clinically relevant bacteria, e.g., *Enterobacteriaceae*, *Salmonella*, *Shigella* or *Serratia* [23,40,41].

Integrons contribute to the horizontal gene transfer (HGT) of ARGs, since they often acquire ARG-encoding gene cassettes that can be mobilized by the encoded integrase at the recombination sites (*attC*). Excised gene cassettes may be inserted into other integrons at their recombination sites (*attI*). Whereas chromosomal integrons sometimes carry hundreds of gene cassettes of diverse or unknown functions, mobile and clinically relevant class 1 integrons usually harbor a maximum of six gene cassettes, mostly encoding ARGs [42]. The insertion and excision of genes are mediated by a tyrosine recombinase, the integron-integrase encoded by the *intI* gene. Class 1 integron-integrases are primarily associated with clinical bacteria [42]. The respective *intI1* gene is therefore used as a marker gene for anthropogenic pollution [42,43,44].

Plasmids of the IncP-1 group have a broad host range and are self-transmissible [45,46]. They have been found in diverse environments such as soil [47,48,49], the rhizosphere [49,50], manure [51], wastewater [52], rivers [53] and estuarine water [54]. IncP-1 plasmids are associated with antibiotic and metal resistance, as well as catabolic functional genes. Being highly efficient at HGT, they are considered important vectors for the dissemination of ARGs [45,55,56]. KorB is involved in global regulation and partitioning of IncP-1 plasmids. The corresponding *korB* gene is highly conserved, and therefore serves as a marker gene for IncP-1 plasmids and HGT [57,58].

We have assessed the fate of the highly prevalent and environmentally important carbapenemase gene *bla_IMP_* [59,60], the ESBL genes *bla_CTX-M_* and *bla_TEM_*, as well as the class 1 integron-integrase gene *intI1* and IncP-1 plasmid regulator gene *korB* as indicators of HGT during the thermophilic composting of human excreta from dry toilets amended with green cuttings, straw, and with and without biochar. Gene-specific TaqMan qPCR was applied to samples taken before, after 14 days, and at the end of composting, after 5 and 5.5 months.

We previously assessed the bacterial community in the same samples using 16S rRNA gene amplicon sequencing, as well as cultivation-based approaches, to analyze potential human pathogens in the mature compost [61]. Moreover, we performed metagenomic sequencing and qPCR, analyzing virulence factors and ARGs conferring resistance to different antibiotic classes [62]. This study complements our previous publications by examining the fate of environmentally relevant beta-lactamase genes and the impact of thermophilic composting on the horizontal spread of resistance genes.

## 2. Materials and Methods

### 2.1. Composting Trial and Sampling

Experimental set-up of the thermophilic composting trial, as well as physico-chemical parameters of the composting substrates were described in detail in [61]. In brief, dry toilet contents, along with green cuttings, straw, and urine, with or without biochar, were composted in windrows in Eckernförde, Germany, between September 2018 and January 2019. The two treatments with (“-B”) and without biochar were set up twice, at an interval of 2 weeks. E1 and E1-B indicate the first, and E2 and E2-B the second repetition of the trial.

Samples were taken on the day of experimental set-up (“start”), after two weeks (“14 days”, only repetition 1), and at the end of the trials after 5.5 (repetition 1) and 5 months (repetition 2), respectively, to compare the initial load of ARGs with the finished product intended to be used in agriculture. The samples after 14 days of composting represent the thermophilic phase [63]. Unfortunately, samples after 14 days of composting were only available from repetition 1.

### 2.2. DNA Extraction and Purification

Total DNA was extracted using the NucleoSpin^®^ Soil gDNA extraction kit (Macherey-Nagel), according to the manufacturer’s instructions. From each sample, 250 mg compost were used, along with lysis buffer SL2 without enhancer. Mechanical disruption was conducted in a FastPrep-24^™^ 5G homogenizer (MP Biomedicals) at 5 m/s for 30 s. A total of 50 µL nuclease-free water was used for DNA elution. The DNA extracts were further purified using a magnetic separation device (NucleoMag^®^ SEP, Macherey-Nagel) and the CleanNGS kit (CleanNA, according to the manufacturer’s instructions. Preliminary tests revealed an optimal dilution of 1:100 for qPCR.

### 2.3. Quantitative RealTime PCR

The qPCR analyses were conducted as stated in [62] on a LightCycler^®^ 480 II instrument (Roche Diagnostics Ltd.). Primer sequences, PCR amplicon sizes, and references are listed in Table 1. The composition of the qPCR mastermix for all *bla* genes and *intI1* was as follows: 1x LightCycler^®^ Probes Master (Roche Diagnostics Germany GmbH), 0.5 µM each primer, 0.1 µM probe, 5 µL template (1:100 dilution of DNA extracts). The total volume per reaction was 20 µL. PCR programs were initiated with a denaturation step at 95 °C for 10 min, followed by 45 cycles of denaturation at 95 °C for 10 s, and annealing/elongation at 60 °C (*bla_CTX-M_*, *bla_IMP_*, *intI1*) or 56 °C (*bla_TEM_*), and 72 °C for 1 s. Standard curves of 10^1^ to 10^8^ copies of purified PCR products were used for quantification. The *korB* qPCR mastermix consisted of 1x LightCycler^®^ Probes Master, 0.4 µM primers -F and -R, 0.2 µM primers -Fz, -Rge, and -Rd, 0.15 µM each probe, 5 µL template (1:100 dilution of DNA extracts), H_2_O ad. 20 µL. The qPCR program was initiated at 95 °C for 5 min, followed by 40 cycles of 95 °C for 15 s; 54 °C for 15 s; and 60 °C, 1 min, including fluorescence measurement. An equimolar mix of plasmids representing different IncP-1 groups was used as a standard for quantification (Table 1). Gene copy-numbers were normalized to the 16S rRNA gene and to 1 g compost. Two-tailed, paired Student’s t-test was conducted to define significant differences between start and end samples of composting. Due to the small number of replicates in the start samples (one mixed sample per compost pile), treatments were combined to test for significance. This means that four values from the start of composting were tested against four values from the end (the mean of the triplicates per pile, respectively). Additionally, the biological replicates of the end samples were analyzed for differences between the two treatments with and without biochar, which means six samples per treatment. This was conducted for both normalizations (to 1 g compost and to the 16S rRNA gene copies). The *p*-values below 0.05 indicate statistical significance, and *p*-values below 0.01 are considered highly significant.

## 3. Results

The present study aimed to analyze the fate of beta-lactamase genes, as well as marker genes for HGT during thermophilic composting of human excreta. These results complement a previous study on the same samples analyzing antibiotic resistance and virulence genes [62].

Figure 1 shows the qPCR results during composting for the ARGs *bla_IMP_*, *bla_CTX-M_* and *bla_TEM_*. No significant changes between start and end samples were observed. For *bla_IMP_*, copy numbers in the final samples show higher variance than copy numbers in the initial samples. *Bla_CTX-M_* and *bla_TEM_* copy numbers exhibit the opposite trend, with higher variances in the start samples (data not shown). The qPCR data for *intI1* and *korB* as marker genes for HGT are shown in Figure 2. Both genes show significant decreases during thermophilic composting. No significant differences were detected between the treatments with biochar (E1-B, E2-B) and without biochar (E1, E2) in the mature compost for any of the genes analyzed (data not shown).

Table 2 depicts the percentage decrease or increase in gene copy numbers during composting. The beta-lactamase genes exhibited mean increases of gene copy numbers from start to end of composting, when changes in % of all treatments were averaged. Only *bla_TEM_* normalized to 1 g compost showed a mean decrease of 42.6%. *IntI1* copy numbers decreased significantly by 90% (normalized to 16S rRNA gene copies) or 95.8% (normalized to 1 g compost) from start to end of composting. *KorB* copy numbers exhibited significant decreases of 99.0% (normalized to 16S rRNA gene copies) and 99.5% (normalized to 1 g compost), respectively.

Table 3 depicts mean gene copy numbers of start and end of composting, irrespective of the treatments with or without biochar. This partly shows different trends, as compared with the calculated decrease or increase shown in Table 2. Mean gene copy numbers of *bla_IMP_* increased during composting, whereas *bla_CTX-M_* decreased slightly and *bla_TEM_* decreased by two orders of magnitude. *IntI1* and *korB* decreased by one and three orders of magnitude, respectively (Table 3). The abundances of *intI1* and *korB* genes were significantly different in both the start and end samples of composting.

## 4. Discussion

The study aimed to assess the fate of three beta-lactam ARGs of high clinical relevance, as well as the class 1 integron-integrase gene *intI1* and *korB* marker gene for IncP-1 plasmids as indicators for HGT during thermophilic composting of human excreta amended with green cuttings and straw, with and without biochar. The trial and results of cultivation-based and molecular analyses on the bacterial community and ARGs for different antibiotic classes have been published previously [61,62].

*Bla_IMP_*, *bla_CTX-M_* and *bla_TEM_* were chosen for the study, since these three genes exhibited increases from start to end of composting in shotgun metagenomic sequencing data on the same samples (data not shown). Using the more sensitive qPCR method, we aimed to quantify this trend. However, the qPCR results showed no clear pattern of changes during composting, with inconsistency in abundances among the replicate samples. This might be due to large variations in the copy numbers of the replicates from the end of composting. Since a sample dilution of 1:100 was applied to avoid inhibition of the qPCR, gene copy numbers in the reactions were very low, in both initial and final samples. This leads to higher variation in replicate measurements [68]. The *bla* gene abundances were also low in comparison with other ARGs analyzed before [62]. This is interesting, since metagenomic data on the same samples revealed the highest abundance for the group of beta-lactam ARGs, with significant decreases during composting [62]. However, the high abundance in the metagenomic data might be due to the broad variety of genes in this group of ARGs, rather than a high prevalence of individual genes [69]. In line with low abundances of the beta-lactamase genes analyzed here, we have not detected *bla_IMP_* or *bla_CTX-M_* through conventional PCR, either [61].

Although IMP-type carbapenemases are distributed worldwide and are endemic in some countries [25,26], their genes have hardly been screened in environmental studies [70,71]. To the best of our knowledge, there are no composting studies that have examined the fate of *bla_IMP_*. A study of the antibiotic resistance profiles of *P. aeruginosa* isolates from different composts and their feedstock materials highlighted the prevalence of carbapenemases. A total of 24% of the isolates showed resistance to carbapenems. Thus, it was one of the most prevalent resistances among the isolates [72]. Carbapenems are used to treat *P. aeruginosa* infections and are therefore often associated with this pathogen [73,74]. Although culture conditions were selective for *P. aeruginosa*, we could only affiliate four isolates out of two hundred from the mature compost to *P. aeruginosa* [61]. Their low prevalence might be one reason for the overall low *bla_IMP_* abundances. The *bla_IMP_* gene is also associated with carbapenem-resistant *Enterobacteriaceae* [25,26]. *Enterobacteriaceae* sequences were found in a 16S rRNA amplicon sequence analysis conducted previously on the same samples with abundances of between 0 and 1.5% of the total bacterial sequences [61]. Shotgun metagenomic sequencing data showed a broader variety of *Enterobacteriaceae* with slightly decreasing relative abundances during composting, from 0.3–0.4% to 0.26–0.27% relative abundances (data not shown). In a recent study by Willms and colleagues, *bla_IMP_* genes were found in grassland and forest soils. Abundances did not differ among different land use strategies. However, a positive correlation with fungal diversity was detected [47]. This might explain rising *bla_IMP_* abundances in the compost, since fungi contribute to composting rise in abundance during the composting process [4].

*Bla_CTX-M_* showed lowest abundances of 3.6 × 10^3^ copies/g out of 18 ARGs analyzed in pig manure composting [75]. The copy numbers were around one order of magnitude lower than in the present trial. A comparatively low abundance of different *bla_CTX-M_* genes seems to be typical for thermophilic composting of animal manures (cattle, cow, chicken, sheep) [18,76,77]. Most studies found a decline in the gene during composting [18,75,76,78]. In sewage sludge and mushroom composting, *bla_CTX-M_* abundance also decreased [79]. Our results do not draw a clear picture, since abundances during the composting process vary for the different treatments and the two composting trials.

*Bla_TEM_* is prevalent in European wastewater-treatment plants [80,81]. Even in the feces of healthy persons without antibiotic treatment, *bla_TEM_* is frequently detected [82,83]. Zhang et al. reported decreases in *bla_TEM_* during sewage sludge composting [79]. Interestingly, the sample peaking in gene abundance (E1-B start) corresponds to the metagenomic sequencing data from the same samples. E1 and E2 exhibited increases during composting, whereas no *bla_TEM_* sequences were detected in E2-B in the metagenomic data (data not shown).

Significant decreases in *intI1* indicate that the composting process reshaped the bacterial community with respect to decreasing the abundance of clinical strains. This corresponds with amplicon and metagenomic sequencing results from the same samples, where decreases in potential human pathogens could be shown [61,62]. In a clinical context, class 1 integron-integrase genes are most common, whereas a broader variety of integron classes is found in the environment [42]. The *intI1* gene serves as a proxy for ARGs, since class 1 integrons frequently carry resistance genes [43]. The significant decline in *intI1* in this study is in line with our previous analyses. Metagenomics, as well as PCR and qPCR results showed decreases in ARGs conferring resistance to different classes of antibiotics [61,62]. Decreases in *intI1* in a study on sewage sludge composting with and without biochar and hyperthermophiles were in the same range as in this work [84]. Liao et al. showed the positive effect of high temperatures on *intI1* removal rates, by comparing hyperthermophilic and thermophilic composting of sewage sludge [85].

The addition of 5 mass-% biochar to sewage sludge compost resulted in *intI1* copy numbers of 4 × 10^5^, as compared with 6 × 10^7^ in the control treatment without biochar [86]. The addition of 2 or 5% biochar increased the removal efficiency of *intI1* gene copies in sewage sludge composting with hyperthermophiles [84]. In the present study, biochar addition of 3–4% did not result in different gene abundances after composting.

Significant decreases in *korB* gene abundances by 99 and 99.5% demonstrated that the composting process can contribute markedly to the decline of IncP-1 plasmids. This group of plasmids is strongly associated with ARGs and HGT [45,55,56]. Their decrease therefore indicates a reduced potential for the dissemination of ARGs in the mature compost. To the best of our knowledge, there are only very few studies investigating IncP-1 plasmids in compost or compost-amended soil. Wu and colleagues demonstrated higher removal of IncP plasmids marker gene copies with higher temperatures during pig manure composting [87]. IncP-1 plasmids have been detected in various environments, often associated with xenobiotic pollution [88]. The recently discovered high diversity of IncP-1 plasmids in the rhizosphere points to the important ecological role of these plasmids [50].

The occurrence of *intI1* gene copies was significantly higher than that of *korB* gene copies in both initial and final samples of composting. This indicates that many of the class 1 integrons are not associated with IncP-1 plasmids. Interestingly, the relative abundance of both genes was previously shown to be correlated with each other in soils subject to mineral fertilization. However, contrary to our data, the concentration of *korB* was higher than *intI1* in the rhizosphere samples [49]. Both *intI1* and *korB* serve as signature genes for the transferability of ARGs via HGT. Our *korB* data show a significant decrease in IncP-1 plasmids during thermophilic composting. Thus, we provide important information on the decreasing potential of HGT due to thermophilic composting.

In conclusion, the beta-lactamase genes *bla_IMP_*, *bla_CTX-M_*, and *bla_TEM_* were detected before and after thermophilic composting of human excreta, in low concentrations. Clear trends in the decrease or increase of copy numbers were not observed. Significant decreases in *intI1* and *korB* genes indicate a decline in clinically relevant bacteria and HGT during composting. This suggests that thermophilic composting can be effective in reducing the public health burden of human excreta.

## Figures and Tables

**Figure 1 microorganisms-11-00308-f001:**
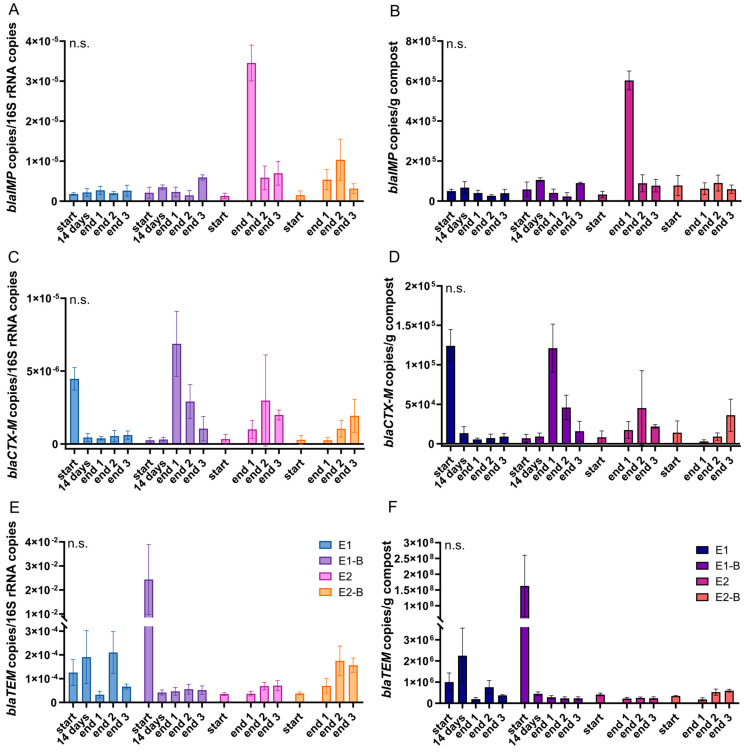
Copy numbers of beta-lactam ARGs *bla_IMP_* (**A**,**B**) and *bla_CTX-M_* (**C**,**D**), as well as *bla_TEM_* (**E**,**F**) as assessed by qPCR. Copy numbers were normalized to 16S rRNA gene copy numbers (left charts) and to 1 g compost (right charts). Bars represent the average of three individual qPCR measurements with standard deviation. Start samples, 14-days samples (only repetition 1) and end samples are depicted individually for the first (E1, E1-B) and second repetition of composting (E2, E2-B). End samples comprise replicates from front (end 1), middle (end 2) and back (end 3) of each compost pile. The *p*-values were calculated to compare start with end samples of composting for all four compost piles combined (average of the triplicate end samples, no differentiation of the treatments). None of the sample sets showed significance (*p* < 0.05). Therefore, they are indicated by “n.s.” (not significant).

**Figure 2 microorganisms-11-00308-f002:**
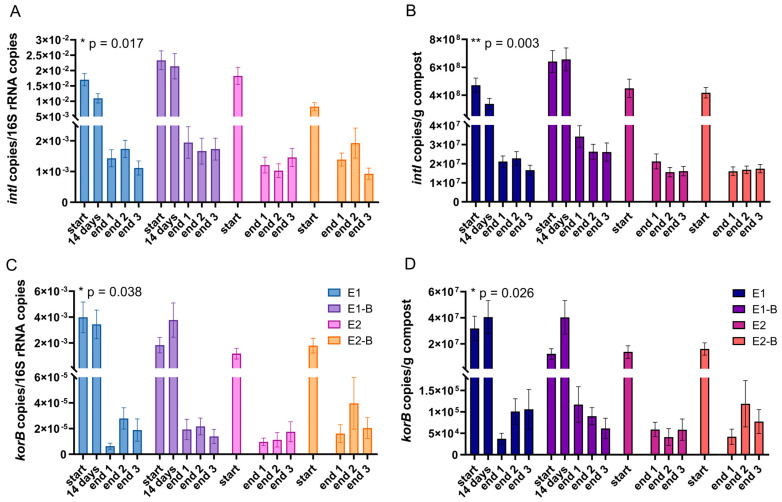
Copy numbers of the class 1 integron-integrase gene *intI1* (**A**,**B**) and IncP-1 marker gene *korB* (**C**,**D**) as assessed by qPCR. Copy numbers were normalized to 16S rRNA gene copy numbers (left charts) and to 1 g compost (right charts). Bars represent the average of three individual qPCR measurements with the standard deviation. Start samples, 14-days samples (only repetition 1) and end samples are depicted individually for the first (E1, E1-B) and second repetition of composting (E2, E2-B). End samples comprise replicates from the front (end 1), middle (end 2) and back (end 3) of each compost pile. Asterisks indicate the *p*-values obtained from Student’s *t*-test (* *p* < 0.05, ** *p* < 0.01) representing the statistical significance of the data. The *p*-values were calculated to compare start with end samples of composting for all four compost piles combined (average of the triplicate end samples, no differentiation of the treatments).

**Table 1 microorganisms-11-00308-t001:** qPCR primers and probes with target gene, oligonucleotide primer sequence, amplicon size, positive control for generation of standards, and primer/probe reference. Standard curves of 10^1^ to 10^8^ gene copies were used for quantification.

Target Gene	Primer/ Probe	Sequence (5′→3′)	Amplicon Size [bp]	Positive Controls	Reference
*bla_CTX-M_*	q_CTXM-F	CAGCTGGGAGACGAAACGTT	63	*E. coli* 1058_16 *	[64]
	q_CTXM-R	CCGGAATGGCGGTGTTTA			
	q_CTXM-P	6FAM- CGTCTCGACCGTACCGAGCCGAC-TAMRA			
*bla_IMP_*	q_blaIMP-F	ATTTTCATAGTGACAGCACGGGC	105	*Klebsiella pneumoniae* R77 IMP-4	[65]
	q_blaIMP-R	CCTTACCGTCTTTTTTAAGCAGCTCATTAG			
	q_blaIMP-P	6FAM- TTCTCAACTCATCCCCACGTATGC-TAMRA			
*bla_TEM_*	q_TEM-F	CACTATTCTCAGAATGACTTGGT	85	*E. coli* 832_16_2 *	[66]
	q_TEM-R	TGCATAATTCTCTTACTGTCATG			
	q_TEM-P	6FAM- CCAGTCACAGAAAAGCATCTTACGG-TAMRA			
*intI1*	q_intI-F	GCCTTGATGTTACCCGAGAG	196	*E. coli* pKJK5 #337	[67]
	q_intI-R	GATCGGTCGAATGCGTGT			
	q_intI-P	6FAM-ATTCCTGGCCGTGGTTCTGGGTTTT-TAMRA			
*korB*	q_korB-F	TCATCGACAACGACTACAACG	118	*E. coli* RP4 (IncP-1α) *E. coli* JM109 pB10 (IncP-1ß) *E. coli* JM109 pQKH545 (IncP-1γ) *E. coli* JM109 pKJK5 (IncP-1ε)	[58]
	q_korB-Fz	TCGTGGATAACGACTACAACG			
	q_korB-R	TTCTTCTTGCCCTTCGCCAG			
	q_korB-Rge	TTYTTCYTGCCCTTGGCCAG			
	q_korB-Rd	TTCTTGACTCCCTTCGCCAG			
	q_korB-P	6FAM- TCAGYTCRTTGCGYTGCAGGTTCTCVAT-TAMRA			
	q_korB-Pgz	6FAM- TSAGGTCGTTGCGTTGCAGGTTYTCAAT-TAMRA			

* clinical isolates from Robert Koch Institute Wernigerode.

**Table 2 microorganisms-11-00308-t002:** Percentage of gene copy numbers at the end of composting, relative to the start copy numbers normalized to the 16S rRNA gene and to 1 g compost.

ARG	Normalization	Decrease from Start to End of Composting [%] *				
		E1	E1-B	E2	E2-B	Mean
*bla_IMP_*	16S rRNA	+36.1	+52.6	+1096.2	+308.2	+373.3
	g compost	−29.5	−12.2	+689.3	−9.5	+159.5
*bla_CTX-M_*	16S rRNA	−88.6	+1323.3	+499.9	+285.8	+505.1
	g compost	−94.1	+773.9	+245.8	+14.1	+234.9
*bla_TEM_*	16S rRNA	−18.1	−99.8	+66.9	+252.5	+50.4
	g compost	−56.0	−99.8	−42.1	+27.4	−42.6
*intI1*	16S rRNA	−91.6	−92.4	−93.2	−82.9	−90.0
	g compost	−95.7	−95.5	−96.1	−96.0	−95.8
*korB*	16S rRNA	−99.6	−99.0	−98.9	−98.6	−99.0
	g compost	−99.7	−99.3	−99.6	−99.5	−99.5

* signs of gene copy numbers were reversed to show increases with positive signs and decreases with negative signs.

**Table 3 microorganisms-11-00308-t003:** Mean copy numbers of ARGs and horizontal gene transfer marker genes irrespective of the treatment with and without biochar, as assessed by qPCR. Copy numbers were normalized to 1 g compost.

ARG	Mean Copy Numbers per Gram Compost	
	Start	End
*bla_IMP_*	5.5 × 10^4^	1.0 × 10^5^
*bla_CTX-M_*	3.8 × 10^4^	2.8 × 10^4^
*bla_TEM_*	4.1 × 10^7^	3.4 × 10^5^
*intI1*	5.0 × 10^8^	2.1 × 10^7^
*korB*	1.9 × 10^7^	7.6 × 10^4^

## Data Availability

Not applicable.

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
