# Peer review of "Fate of Horizontal-Gene-Transfer Markers and Beta-Lactamase Genes during Thermophilic Composting of Human Excreta"

_microorganisms, 2023, doi:10.3390/microorganisms11020308_

Round 1

Reviewer 1 Report

The manuscript no. 2139395 is a appreciable and significant attempt to reduce the spread of the resistance gene in the associated system. The manuscript is drafted well with the proper figures and analysis of result. I have some concern about the experimental design and additional information related to the bacterial community.

1. Authors have provided only the dynamics of a few genes with a single method qPCR.  Please note, composting is a complicated system. The dynamics of the more genes from the related category should be investigated.

2. Second thing, It is very important to see the effect of the factor on entire microbial communities. The data related to the bacterial (or microbial) shifts in the treatment is very important to reach a conclusion. Here, we dont know the status of the bacterial groups (pathogenic/non-pathogenic) in the composting.  I would suggest authors if they can provide a bacterial community structure from the important time points, would be better.

In addition to the above comments, there is no data is provided related to the physic-chemical properties of the compost. What was the change in pH and nutrients? 

The upgradation of the MS with the abovementioned points will improve the strength of the study.

Author Response

Dear Reviewer,

Thank you for the critical reading of our manuscript and your thoughtful comments that we would like to address step by step in the following. 

  1. Authors have provided only the dynamics of a few genes with a single method qPCR.  Please note, composting is a complicated system. The dynamics of the more genes from the related category should be investigated.

Answer: Thank you for your suggestion. We have already published two papers on the same composting trial analyzed here. Several ARGs conferring resistance to different antibiotic classes (beta-lactams, chloramphenicol, erythromycin, fluoroquinolones, gentamicin, kanamycin, sulfamethoxazole, tetracycline, vancomycin) were analyzed both by PCR (ampC, blaZ, blaKPC, blaIMP, blaNDM, blaOXA-48,blaCTX-M, mecA, catpIP501, catLM, ermA, ermB, ermC, ermD, ermG, qnrA, qnrB, aac6-aph2, aph(2)-Ib, aph(2)-Ic, aadDpSK41, aph3-III, sul1, sul2, tetA, tetB, tetKpT181, tetL, tetM, tetS, vanB, vanC1/C2) and qPCR (aph(2’’)-Ia, aph(3’)-IIIa, ermA, ermB, sul1, sul2, tetL and tetS) (DOI: 10.3389/fmicb.2022.824834, DOI: 10.3389/fmicb.2022.826071 ). We have clarified this in our manuscript in lines 102-103.

  1. Second thing, It is very important to see the effect of the factor on entire microbial communities. The data related to the bacterial (or microbial) shifts in the treatment is very important to reach a conclusion. Here, we dont know the status of the bacterial groups (pathogenic/non-pathogenic) in the composting.  I would suggest authors if they can provide a bacterial community structure from the important time points, would be better.

Answer: Thank you for pointing this out. We agree. In our previous manuscript DOI: 10.3389/fmicb.2022.824834 we have conducted amplicon sequencing of the 16S rRNA gene and thereby analyzed the bacterial community before and after composting with a focus on human pathogenic bacteria. The abundance of potential pathogens was shown to decrease during composting. Cultivation-based techniques were used, as well, for the isolation of putative pathogens in the mature compost. Except for one sample, bacterial indicators in the mature compost were below threshold levels for organic fertilizer. Most serious human pathogens that were targeted during isolation from the mature compost could not be identified. In addition, shotgun metagenomics on the same samples were published in DOI: 10.3389/fmicb.2022.826071, focusing on virulence factors and the microbial community. Virulence factors (bacterial motility genes, biofilm formation genes, toxin production genes) decreased during composting. We emphasized this in the manuscript in lines 102-106, 210-212, and 266-268.

  1. In addition to the above comments, there is no data is provided related to the physic-chemical properties of the compost. What was the change in pH and nutrients?

Answer: Details on the experimental set-up including physico-chemical properties of the compost were published in our paper DOI: 10.3389/fmicb.2022.824834. These include the initial composition of substrates, as well as their dry matter, volatile solids, and C/N-ratio. Temperature curves and data, as well as moisture contents are shown, as well. pH values were exemplarily determined for one repetition of the composting trial. pH ranged from 7.06 to 7.0 (start to end of composting) with increases to 8.52 after 42 days of composting (data not shown). We included the reference to the paper in lines 109-110.

We hope that we could clarify the issues.

Kind regards,

Katharina Werner

Reviewer 2 Report

Few data regarding beta-lactamase genes and horizontal gene transfer marker genes in thermophilic composting of human excreta have been reported so far. I found this topic of scientific interest, since it alerts for the role of this kind of organic waste might have on disseminating antimicrobial resistance, especially because it is used in agriculture and could contaminate the environment and enter the food chain. Furthermore, this kind of surveillance is important to develop strategies to address the drug resistance problem. The manuscript is well written and the results will contribute to the current knowledge in the field.

The message “Error! Reference source not found.” Appears in almost all reference in the Results section, please check the references.

Line 39: delete the word specially

Author Response

Dear Reviewer,

Thanks for the critical reading of the manuscript. We appreciate your comments that we address in the following.

1. The message “Error! Reference source not found.” Appears in almost all reference in the Results section, please check the references.

Answer: Thank you for your comment. We have updated the cross-links, that should be displayed correctly now.

2. Line 39: delete the word specially

Answer: We deleted the word.

Kind regards,

Katharina Werner

Reviewer 3 Report

This manuscript, “Fate of horizontal gene transfer markers and beta-lactamase genes during thermophilic composting of human excreta”, aimed to discusses the impact of thermophilic composting on ARGs and have shown varying levels of effectiveness in reducing the presence of these genes. Overall, the manuscript is well-written. However, as the author mentioned in the manuscript, this study used the identical design, samples and methods as some previous publications by the same group (doi: 10.3389/fmicb.2022.824834, doi: 10.3389/fmicb.2022.826071). Despite the different ARGs reported in these publications, I failed to see the novelty and necessity of publish yet another paper with a similar conclusion. The quality of the manuscript would be improved if the authors can carefully explain what new insights this manuscript provided compared to the previous publications.

The following comments or suggestions, if can be addressed, would further strengthen this manuscript.

  1. Both samples with and without biochar were analyzed in the study. Although mentioned in the discussion section (Line 272-276), it might be helpful if the authors can briefly describe/review the effect of biochar on ARG removal in the introduction section.
  2. Line 111-113: Samples were collected at several different time points. Can the authors provide some explanations on why these specific time points were chosen. And why 14 days were not included in the second repetition.
  3. A more detailed explanation of the statistical methods is needed. If I understand it correctly, authors combined E1, E1-B, E2, and E2-B samples together, and also combined the triplicate end samples together. If that is the case, considering the large variations between different compost piles, and also the large variations between the triplicate end samples, authors need to justify why it is OK to combine all the samples. The results can be largely biased due to outliers (e.g., blaIMP g compost results) which is a major limitation of the current statistical method used.
  4. The presentation of the results needs to be improved. Many of the numbers/results mentioned in the results section were either not shown or cannot be directly found in the table/figures.
  5. Numbers in Table 2 is counter intuitive. It may be more reader-friendly if the signs can be reversed, i.e., minus implies decrease.
  6. As mentioned above, in the discussion section, the authors should carefully explain what new insights this manuscript provided compared to the previous publications.

Author Response

Dear Reviewer,

Thank you for the critical reading of our manuscript and your thoughtful comments that we would like to address step by step in the following. 

1. Both samples with and without biochar were analyzed in the study. Although mentioned in the discussion section (Line 272-276), it might be helpful if the authors can briefly describe/review the effect of biochar on ARG removal in the introduction section.

Answer: Thank you for your comment. The effect of biochar on the composting process was briefly added in lines 36-37 and 49-56 (lines numbers for version with track changes).

2. Line 111-113: Samples were collected at several different time points. Can the authors provide some explanations on why these specific time points were chosen. And why 14 days were not included in the second repetition.

Answer: We added some information on the sample set to section 2.1, lines 128-131. Unfortunately, samples after 14 days were not available from repetition 2.

3. A more detailed explanation of the statistical methods is needed. If I understand it correctly, authors combined E1, E1-B, E2, and E2-B samples together, and also combined the triplicate end samples together. If that is the case, considering the large variations between different compost piles, and also the large variations between the triplicate end samples, authors need to justify why it is OK to combine all the samples. The results can be largely biased due to outliers (e.g., blaIMP g compost results) which is a major limitation of the current statistical method used.

Answer: Thank you very much for pointing this out. We tried to clarify this in section 2.3, lines 158-166. Regarding the blaIMP outlier, we conducted the t-test excluding this value. The result is still not significant. Besides, the qPCR result of the respective blaIMP outlier (E2, end 1) was reproducible for the technical replicates, as shown through the standard deviation in the figure (Figure 1A and 1B).

4. The presentation of the results needs to be improved. Many of the numbers/results mentioned in the results section were either not shown or cannot be directly found in the table/figures.

Answer: Thank you for your comment. We see that it can be misleading that the mean copy numbers of the genes analyzed are not directly apparent from the figures. We therefore added another table listing the mean copy numbers of the genes analyzed in start and end samples of composting (Table 3, lines 230-233). The results section was moreover thoroughly modified and restructured to present our results better understandable for the reader (176-233).

5. Numbers in Table 2 is counter intuitive. It may be more reader-friendly if the signs can be reversed, i.e., minus implies decrease.

Answer: Thank you for the valuable comment. Signs of gene copy numbers were changed to make it more reader friendly. This is now explained in the footnote of Table 2: “signs of gene copy numbers were reversed to show increases by positive signs and decreases by negative signs” (lines 220-221).

6. As mentioned above, in the discussion section, the authors should carefully explain what new insights this manuscript provided compared to the previous publications.

Answer: In our previous publications, we did neither assess the abundance of beta-lactamase genes using qPCR, nor analyze the potential of ARG transfer before, during and at the end of the composting process. To the best of our knowledge, there are no composting studies that have examined the fate of blaIMP. Moreover, the present study adds important information on the presence of class 1 integrons and IncP-1 plasmids in compost. The first is of high clinical relevance regarding the transfer of antibiotic resistance. The IncP-1 group of plasmids is environmentally important, has large HGT potential with high plasmid transfer rates among a large variety of Gram-negative bacteria. To the best of our knowledge, there are only very few studies investigating IncP-1 plasmids in compost or compost-amended soil. Assessing the potential risk of compost from dry toilet contents, the quantity and decrease of these HGT marker genes is an important indicator for the efficiency of the composting system. We tried to clarify this in the manuscript in lines 111-117 and 327-330.

Please find attached the revised manuscript with track changes.

Kind regards,

Katharina Werner

Reviewer 4 Report

Dear authors,

It is a pleasure to evaluate your manuscript. The fate of horizontal gene transfer markers and beta-lactamase genes during thermophilic composting of human excrement is relevant because it can provide insight into the effectiveness of this process in reducing the public health risk associated with human excrement. Beta-lactamase genes are often used as markers for horizontal gene transfer, as they can be easily transferred between bacterial species. By studying the presence and concentration of these genes during thermophilic composting, researchers can determine whether this process effectively reduces the risk of horizontal gene transfer, which is the transfer of genetic material between organisms that does not occur through reproduction. This information is essential for understanding how best to manage and treat human manure to minimize the potential for transmission of harmful pathogens.

Although the conclusion is not within a distinct section is well-structured, according to the journal guidelines.

Line 36: change [3], [4] to [3,4]

Line 37: change [4]–[6] to [4–6]

Do these changes throughout the document.

Line 36: start a new paragraph

Line 116: do not start a period with a number. Either you change the word order or write the number in full.

Lines 139: state clearly which variable the student's t-test is comparing and what the test outcomes mean.

Line 153 (and others): review the references to figures and tables. You probably did not update these references when you copied the content from another document.

Figure 1 (also applies to 2): you could organize the page in "portrait" format (or vertical) and reorganize the side of the graph by the side. If you do that, you do not have to skip from A to D, B to E, etc. For instance, ARGs blaIMP could become A and B, and so on. 

Still, in the figure 1 caption, you mention non-existing asterisks. Is it not easier to say that no observation showed significant p < 0.05? Figure 2 caption seems fine because you presented the values.

Regarding your conclusion, it is comprehensive and coherent with your observations. Thermophilic composting of human excreta showed a decrease in the presence of clinically relevant bacteria and horizontal gene transfer. Beta-lactamase genes were detected in low concentrations before and after the process, but no apparent increase or decrease trends were observed. This suggests that thermophilic composting may effectively reduce the public health risk associated with human excrement.

Yours sincerely

Author Response

Dear Reviewer,

Thank you for the critical reading of our manuscript and your thoughtful comments that we would like to address step by step in the following. 

1. Although the conclusion is not within a distinct section is well-structured, according to the journal guidelines.

Answer: Thank you for your comment. As this is a short communication, we left the conclusion as part of the discussion.

2. Line 36: change [3], [4] to [3,4]

Line 37: change [4]–[6] to [4–6]

Do these changes throughout the document.

Answer: Thank you for your comment. We changed the references according to the journal guidelines.

3. Line 36: start a new paragraph

Answer: We started a new paragraph.

4. Line 116: do not start a period with a number. Either you change the word order or write the number in full.

Answer: Thank you for the careful reading of our manuscript. We rephrased the sentence, so that it does not start with a number.

5. Lines 139: state clearly which variable the student's t-test is comparing and what the test outcomes mean.

Answer: Thank you for pointing this out. We tried to clarify the method by adding more information to chapter 2.3, lines 158-166 (line numbers for the version with track changes).

6. Line 153 (and others): review the references to figures and tables. You probably did not update these references when you copied the content from another document.

Answer: Thank you for your important comment. We corrected the false cross-references.

7. Figure 1 (also applies to 2): you could organize the page in "portrait" format (or vertical) and reorganize the side of the graph by the side. If you do that, you do not have to skip from A to D, B to E, etc. For instance, ARGs blaIMP could become A and B, and so on. 

Answer: We have reorganized the graphs in Figures 1 and 2 accordingly.

8. Still, in the figure 1 caption, you mention non-existing asterisks. Is it not easier to say that no observation showed significant p < 0.05? Figure 2 caption seems fine because you presented the values.

Answer: Thank you for this important comment. We changed the figure caption accordingly.

Please find attached the revised version of the manuscript with track changes.

Kind regards,

Katharina Werner

Round 2

Reviewer 1 Report

The manuscript is improved with the relevant description. I recommend this article in the present form to be published in microorganisms.

Reviewer 3 Report

The author has adequately addressed all comments and there are no further questions. Thank you.